# Recent Developments in Pharmacotherapy of Depression: Bench to Bedside

**DOI:** 10.3390/jpm13050773

**Published:** 2023-04-29

**Authors:** Mujeeb U. Shad

**Affiliations:** 1Valley Health System (VHS), Las Vegas, NV 89118, USA; mujeebushad@gmail.com; 2The Department of Psychiatry, University of Nevada, Las Vegas, School of Medicine, The Touro University of Nevada College of Osteopathic Medicine (TUNCOM), Henderson, NV 89014, USA; 3The University of Nevada, Las Vegas, NV 89154, USA

**Keywords:** developments, pharmacotherapy, depression, bench, bedside

## Abstract

For the last 70 years, we did not move beyond the monoamine hypothesis of depression until the approval of the S-enantiomer of ketamine, an N-methyl-D-aspartate (NMDA) receptor blocker and the first non-monoaminergic antidepressant characterized by rapid antidepressant and antisuicidal effects. A similar profile has been reported with another NMDA receptor antagonist, dextromethorphan, which has also been approved to manage depression in combination with bupropion. More recently, the approval of a positive allosteric modulator of GABA-A receptors, brexanolone, has added to the list of recent breakthroughs with the relatively rapid onset of antidepressant efficacy. However, multiple factors have compromised the clinical utility of these exciting discoveries in the general population, including high drug acquisition costs, mandatory monitoring requirements, parenteral drug administration, lack of insurance coverage, indirect COVID-19 effects on healthcare systems, and training gaps in psychopharmacology. This narrative review aims to analyze the clinical pharmacology of recently approved antidepressants and discuss potential barriers to the bench-to-bedside transfer of knowledge and clinical application of exciting recent discoveries. Overall, clinically meaningful advances in the treatment of depression have not reached a large proportion of depressed patients, including those with treatment-resistant depression, who might benefit the most from the novel antidepressants.

## 1. Introduction

For the last seven decades, clinical practice has focused on developing “me too” antidepressants based on different molecular targets within three major monoamine neurotransmitter systems, serotonin, epinephrine, and dopamine. Despite the differences in types and nature of adverse effects, monoaminergic antidepressants do not differ significantly in their antidepressant efficacy, regardless of the molecular target(s). In addition, these antidepressants have not been adequately effective at managing treatment-resistant depression (TRD). Most importantly, all monoaminergic antidepressants require a period of 4–6 weeks for optimal efficacy. Therefore, a severely depressed patient has to wait for a response during the initial stages of depression when urgent treatment is required to prevent suicidality. Still, there is no guarantee that the first antidepressant will be effective after waiting 4–6 weeks.

The recent paradigm shift from monoamine to glutamatergic and gamma amino butyric acid (GABA)-based hypotheses [1,2] has successfully addressed some unmet needs in treating depression. Although the central role of the most abundant neurotransmitters, glutamate and GABA, in brain function has been known for a long time, it took us several decades to develop antidepressants directly affecting these neurotransmitters. Ketamine is an N-methyl-D-aspartate (NMDA) receptor blocker, indirectly resulting in increased levels of brain-derived neurotrophic factor (BDNF) and rapid onset synaptic neuroplasticity responsible for the quicker onset of antidepressant and antisuicidal effects. Although ketamine was first discovered 60 years ago, it was not until 2000 that the antidepressant effects of ketamine were found serendipitously. However, the rapid onset of ketamine’s antidepressant and antisuicidal effects fueled the formal research that provided strong support for the novel glutamatergic hypothesis of depression [1]. Ketamine has not yet been investigated in preclinical trials but is already used as a 45-minute intravenous infusion to manage acute suicidality in emergency rooms nationwide. Although the rapid antidepressant effects of ketamine could not be translated into the bench-to-bedside transfer of knowledge for the general population, it paved the way for preclinical trials with the intranasal administration of the S-enantiomer of ketamine for its approval to be used in patients with treatment-refractory depression (TRD) [3].

Another exciting development was the FDA approval of brexanolone (a GABA modulator and an analog of allopregnanolone) as the first effective treatment for postpartum depression (PPD), which strongly supports the GABAergic hypothesis [2]. However, brexanolone is an expensive 60-h infusion and, like esketamine, requires registration with the REMS for mandatory monitoring. Although these novel developments are exciting and could be lifesaving, a large number of patients do not have access to these expensive treatments, impairing the transfer of knowledge from the bench to bedside. Therefore, the general population may not benefit from recent psychopharmacological advancements without checks and balances on new drug costs. Nevertheless, mandatory monitoring and the extremely high costs associated with ketamine and esketamine in treating depression have been somewhat addressed after the FDA recently approved another N-methyl-D-aspartate (NMDA) receptor blocker, dextromethorphan, in combination with an older antidepressant, bupropion [4]. Unlike ketamine and esketamine, dextromethorphan–bupropion does not require mandatory registration or monitoring. In addition, dextromethorphan–bupropion is approved for nonrefractory depression, which means that there is a broader clinical application of bench-to-bedside research with dextromethorphan–bupropion than with ketamine or esketamine. These differences could be due to the better tolerability and safety of the dextromethorphan–bupropion combination than that of ketamine or esketamine, with bupropion being already approved as an antidepressant. Along the same lines, zuranolone, an orally administered alternative to the expensive and parenteral brexanolone, is currently in the final stages of development as an antidepressant and potentially as the treatment for PPD [5,6]. Although approval of dextromethorphan–bupropion and future approval of zuranolone may mitigate some of the cost and monitoring barriers, it is still possible that most patients may not benefit from these antidepressants due to a lack of insurance approval. Even in patients who can afford these expensive medications, mental healthcare providers and trainees may be reluctant to change their comfortable prescribing practices to try novel antidepressants unless they are properly educated and trained. In addition, our healthcare systems have not completely recovered from the negative effects of the COVID-19 pandemic, and patients may have become more cautious in trusting and adopting novel strategies [7,8].

The main aim of this review paper is to provide a conceptual framework to understand paradigm shifts from the traditional monoamine to novel glutamatergic and GABAergic hypotheses of depression to facilitate the clinical application of knowledge from the bench to bedside. This update incorporates a mechanism of action-based classification of newly approved antidepressants to facilitate the optimal use of these medications whenever possible. The provided information allows clinicians to select novel antidepressants safely and effectively, particularly in patients with treatment-refractory depression (TRD), who are known to have an increased vulnerability to adverse effects.

The following paragraphs provide an update on novel recently approved antidepressants based on their mechanism(s) of action-based classification, including relatively new monoamine antidepressants, vilazodone, a selective partial agonist and reuptake inhibitor (SPARI), levomilnacipran, the latest and unique addition to the serotonin–norepinephrine reuptake inhibitors (SNRIs), and vortioxetine, a multimodal antidepressant (MMA).

## 2. Selective Partial Agonist and Reuptake Inhibitor (SPARI)—Vilazodone

Vilazodone was FDA-approved in 2011 as the first selective partial agonist and reuptake inhibitor (SPARI). However, vilazodone has much higher affinity for 5HT1A as a partial agonist than the serotonin reuptake pump to be labeled as another SSRI [9]. As a partial agonist for the presynaptic 5HT1A autoreceptors, vilazodone can potentially downregulate these autoreceptors. The downregulation of autoreceptors results in an increased release of serotonin presynaptically, which is thought to mediate antidepressant effects [10]. This mechanism is opposite to that of an SSRI, which initially stimulates 5HTIA autoreceptors to reduce the presynaptic release of serotonin [11] followed by a downregulation period of over 4–6 weeks that is putatively required for antidepressant effects [11,12]. Thus, faster autoreceptor downregulation may lead to a more rapid onset of therapeutic efficacy (Table 1). Clinicians have the option to add a partial agonist, buspirone, to create their own vilazodone-like profile if their patients are already on an SSRI with reducing antidepressant response. Another mechanism with which to speed up the antidepressant response is the addition of a β-adrenergic receptor antagonist, pindolol, which is a 5-HT_1A_ antagonist and does not need any downregulation to achieve a faster onset of antidepressant effect [11,13,14].

Another benefit of pharmacotherapy with a 5-HT_1A_ partial agonist could be the therapeutic effects for anxiety symptoms supported by the approval of buspirone in treating generalized anxiety disorder. Furthermore, 5HT1A partial agonism may reduce or prevent gastrointestinal upset and sexual dysfunction frequently reported with SSRIs [15] (Table 1). However, there is a lack of evidence to support the efficacy of monotherapy with 5-HT_1A_ agonists in major depression [16]. Even augmentation with buspirone, a partial agonist at the 5HT1A receptors, has failed to enhance remission with up to 14 weeks of citalopram treatment [17]. However, pindolol augmentation of SSRIs produced positive effects in terms of the faster onset of action and symptom resolution at the endpoint [18,19], but not in all studies [20,21,22]. It is possible that the contradictory results from these studies could be due to differences in the way the SSRI was augmented, the study population, the study design, and the pindolol dose [22]. Howeer, the majority of positive pindolol augmentation studies were conducted with either fluoxetine [23,24,25,26] or paroxetine [27,28,29,30] because both these SSRIs increase plasma levels of pindolol, supporting the earlier finding that higher levels of pindolol may be more effective [13].

The clinical efficacy of vilazodone was demonstrated in two large randomized, double-blind, placebo-controlled trials [31,32] on depressed adults. In the first study [31], significant improvements in MADRS and HDRS-17 scores were reported in the vilazodone group as early as in week 1, suggesting a rapid onset of action. However, in the second trial [32], a significant difference in MADRS scores was observed after six weeks of treatment. The most commonly reported placebo-adjusted rates of adverse events reported at ≥5% include diarrhea (19%), and nausea (18%) [32,33]. The gastrointestinal adverse effects are high enough to recommend a 2-week titration to 40mg/day of vilazodone to improve medication adherence. However, this 2-week titration may take away the advantage of the rapid onset of efficacy with vilazodone resembling that observed with SSRIs or SNRIs. However, sexual dysfunction was minimal over the 8-week trial duration in the vilazodone group and appeared to be similar to the rates in the placebo group [33]. In addition, vilazodone has been reported to cause less emotional blunting than SSRIs or SNRIs do [34].

Since vilazodone is metabolized by CYP3A4 [35], drugs that are inhibitors of CYP3A4, such as ketoconazole, can increase vilazodone concentrations by 50%, requiring starting dose reductions of 50% when vilazodone is concomitantly used with a drug such as ketoconazole [35]. In contrast, an inducer of CYP3A4, such as carbamazepine, may result in a 45% decrease in vilazodone, thus requiring a higher dose of vilazodone to be effective [36,37,38]. Vilazodone does not significantly induce or inhibit any CYP enzymes [36]. Genetic polymorphisms in CYP3A4 activity are minimal and substantially less clinically relevant than drug–drug interactions are [35].

In general, the faster onset of an antidepressant response [32], low sexual dysfunction, and reduced emotional blunting underscores the clinical utility of vilazodone, particularly in patients developing significant sexual adverse effects and/or initial anxiogenic effects with SSRIs.

## 3. Serotonin Norepinephrine Reuptake Inhibitor (SNRI)—Levomilnacipran

Levomilnacipran was approved in 2013 as a selective-norepinephrine reuptake inhibitor (SNRI). However, levomilnacipran is the only SNRI with a two-fold higher potency for NET than that of SET [39,40] and an over 15-fold higher selectivity for NET than that of duloxetine, desvenlafaxine, or venlafaxine [40]. Levomilnacipran is the levorotary enantiomer of its precursor, milnacipran, which is only approved for fibromyalgia in the United States, and levomilnacipran is ten times more potent for norepinephrine and serotonin transporters than the dextrorotary form is. [40,41,42] (Table 1) In addition, levomilnacipran does not have any significant direct receptor effects on monoamines or electrolyte channels [40].

Interestingly, all SNRIs have been effective at treating fibromyalgia, and one of them, duloxetine, has been FDA-approved for this indication [43,44]. However, it is yet to be seen if a more potent inhibition of NET will be more effective in managing fibromyalgia or other pain syndromes than other SNRIs that have more potent inhibition of SET. Nevertheless, a significant improvement when taking levomilnacipran was reported on a disability scale used in phase III preclinical trials, suggesting that the noradrenergic component may not only contribute to the antidepressant response but also reduce fatigue and enhance functionality [45]. Levomilnacipran is reported to inhibit >90% of norepinephrine reuptake and >80% of serotonin at or above a daily dose of 40 mg [40]. Perhaps due to its norepinephrine effects, levomilnacipran may be more effective for fatigue, which was reported to be a persistent problem in about 60% of the patients despite continuous treatment with SSRIs in the STAR*D study [46]. Noradrenergic mechanisms may also explain the most significant drug–placebo differences in response rates in the group aged ≥60 years [45]. This finding is contrary to earlier observations that older people may not have as good an antidepressant effect compared with younger depressed populations [47]. Moreover, there were minimal gender differences in the antidepressant response to levomilnacipran [48], which supports the notion that antidepressants with more potent serotonin effects, such as SSRIs, may have a better response in females, especially in premenopausal females due to the synergistic impacts between serotonin and estrogen [49,50]. Levomilnacipran also did not differentiate in terms of antidepressant response based on the severity of depression, which is interesting as few studies have suggested that antidepressants may be less effective in severe depression [51]. Similar results were observed with another SNRI, desvenlafaxine [46].

Pharmacokinetically, levomilnacipran has a relatively short half-life, but an extended-release formulation allows once-daily dosing between 40–120 mg/day with gradual and slow titration based on antidepressant response and tolerability [52] (Table 1). In general, the higher doses of levomilnacipran (i.e., between 80 to 120 mg/day) have been more effective than the lower doses have (i.e., around 40 mg/day) [52]. This is perhaps because higher doses of levomilnacipran may allow a more significant blockade of the SET to contribute to dual-mechanism-action efficacy. The same is true for venlafaxine, which only incorporates norepinephrine effects at higher doses [53]. The most common treatment-emergent adverse effects support the more prominent noradrenergic effects of levomilnacipran, including the placebo-adjusted frequency of 11% for nausea, 4% for headache, 6% for constipation, 5% for erectile dysfunction, 3% for dry mouth, 4% for tachycardia, and 7% for sweating [45]. Clinically serious adverse effects are rare but include hypertension, chest pain, and deep venous thrombosis, which could also be related to the more potent noradrenergic effects. Therefore, blood pressure monitoring is even more critical with levomilnacipran than with any other SNRI.

Levomilnacipran is primarily metabolized via the CYP3A4 isoenzyme. In vitro, studies have observed interactions with potent CYP3A4 inhibitors, such as ketoconazole, clarithromycin, and ritonavir (Table 1). A recent in vivo study showed a significant increase in levomilnacipran concentrations when it was coadministered with ketoconazole [54]. However, the study did not show a significant decrease in levomilnacipran concentrations when it was coadministered with the strong CYP3A4 inducer carbamazepine [54]. Although the evidence is limited, the manufacturer advises using a lower dosage of levomilnacipran with strong CYP3A4 inhibitors, specifically 80 mg daily or less [55]. In most cases, a dosage decrease would not be warranted if the CYP3A4 inhibitor is used short-term, such as in the case of a short course of ketoconazole as an antifungal. Still, symptoms of levomilnacipran toxicity, including tachycardia and hypertension, should be monitored [55]. In addition, despite the lack of evidence for carbamazepine-induced rapid clearance, levomilnacipran requires closer monitoring for a decrease in or loss of efficacy with strong CYP3A4 inducers [54]. Levomilnacipran is not known to induce or inhibit any CYP enzymes significantly. In addition, levomilnacipran, like other SNRIs, can be involved in drug interactions if concomitantly given with inhibitors or inducers of CYP3A4 [54]. Like vilazodone, genetic polymorphisms in CYP3A4 activity are minimal and usually not as clinically relevant as drug–drug interactions involving CYP3A4 are [1] (Table 1).

## 4. Multimodal Antidepressants (MMAs)—Vortioxetine

Vortioxetine was also approved in 2013 as the first representative from the multimodal antidepressant (MMA) class [56]. However, many other antidepressants can qualify for multimodality for their clinical utility, such as mirtazapine, which has multiple desirable targets, including alpha-2 adrenergic autoreceptor and heteroreceptor blockade, resulting in increased serotonin and norepinephrine effects, respectively, along with 5HT2A, 5HT2C, and 5HT3 blockade [57] (Table 1).The clinical benefits of 5HT2C and H1 receptor blockade by mirtazapine have already been reported as uniquely beneficial in medically frail older nursing home patients not eating and sleeping well [57]. However, the sedative effects of mirtazapine are stronger at lower doses [57]. Vortioxetine multimodal effects involve 5-HT3, 5-HT7, and 5-HT1D receptor antagonism, a 5-HT1B receptor partial agonism, and a 5-HT1A receptor agonism, along with SERT inhibition [58,59,60]. Like mirtazapine, these multimodal effects of vortioxetine may add to its clinical profile. Vortioxetine is the only antidepressant that is FDA-approved for its procognitive effects on motor speed, attention, and visuoperceptual functions, which are independent of its antidepressant effects [61,62,63]. Although the actual neurobiological mechanisms underlying the procognitive effects of vortioxetine are not completely understood, it is postulated that differential effects of vortioxetine on 5HT1 receptors (i.e., 5HT1A receptor agonism, 5HT1B partial agonism, and 5HT1D antagonism) with checks and balances on serotonin neurotransmission may be contributory [64]. This symphony of vortioxetine’s effects on differentially located 5HT1 receptors has also been proposed to mediate its antidepressant effects with only a 40–50% blockade of the SET at the minimum effective dose of 5 mg/day instead of the 60–80% SET blockade required for SSRIs and SNRIs [65]. However, the preclinical trials found better antidepressant responses with relatively higher doses of 10–20 mg/day. The desirable effect of 5HT3 receptor blockade is the reduction in GI adverse effects (i.e., nausea, vomiting, and diarrhea) and the reduction in the GABA-mediated activation of interneurons, thereby increasing serotonin levels [64]. Vortioxetine also blocks the SET to enhance serotonin activity, like SSRIs. Last but not least are the beneficial effects of vortioxetine in blocking 5HT7 receptors, which in addition to the differential effects on 5HT1 receptors, may mediate some of vortioxetine’s procognitive effects in depressed patients [66,67]. Finally, through its multimodal mechanism, vortioxetine enhances neurotransmitters such as dopamine, histamine, noradrenaline, and acetylcholine [64]. These multimodal effects may all contribute to the vortioxetine-induced increase in hippocampal neurogenesis associated with the antidepressant effects [68].

Pharmacokinetically, vortioxetine has a linear relationship between its dose and plasma levels, which increases the predictability of the dose-dependent response and adverse effects. In addition, vortioxetine has a long half-life of up to 70 h, which means that vortioxetine can be administered once a day, and despite it having no biologically active metabolites, there is a low risk of discontinuation symptoms even after abrupt discontinuation [69]. However, like many other antidepressants, vortioxetine may also be involved in drug interactions.

Vortioxetine is primarily metabolized by CYP2D6 to an inactive metabolite with minimal contributions from other CYP enzymes, reducing the risk of clinically meaningful interactions except for drugs that are substrates or inhibitors of CYP2D6 [70] (Table 1). Concurrent use of CYP2D6 inhibitors, such as bupropion, fluoxetine, or paroxetine, can significantly increase the plasma levels of vortioxetine [70,71] A significant increase in vortioxetine levels with bupropion can result in nausea, vomiting, insomnia, and dizziness, requiring a significant dose reduction. In contrast, rifampin can significantly reduce vortioxetine’s peak concentration by 51% and its area under the curve by 75% [70]. Thus, a dose increase may be required if vortioxetine is used with CYP inducers, such as carbamazepine, phenytoin, rifampin, and barbiturates [70]. However, the enzyme’s induction is most likely mediated by enzymes other than CYP2D6, including CYP3A4. In addition, genetic polymorphisms in CYP2D6 activity might also be clinically relevant. For example, an increase in vortioxetine dose may be required in ultra-rapid metabolizers for CYP2D6 and a decrease in poor metabolizers for CYP2D6 [2]. Vortioxetine is not known to induce or inhibit any CYP enzymes [70].

Vortioxetine has been linked with an increased risk of bleeding, as is the case for all serotonin transporter blockers. The increased bleeding risk is due to the blockade of serotonin transporters in platelets that are similar those in the neurons [72]. The common adverse effects of vortioxetine include nausea, headache, diarrhea, and dry mouth [65]. In addition, vortioxetine has been associated with sexual dysfunction, which is the case with serotonergic antidepressants, such as SSRIs [62]. However, interestingly, sexual dysfunction was reported to be higher with subtherapeutic doses (i.e., 2.5 mg/day) than with 5 mg/day [73]. Vortioxetine was not associated with a clinically significant increase in suicidal ideation and behavior [74]. In addition, vortioxetine did not significantly affect cardiovascular or metabolic measures, weight, or QTc effects during preclinical trials [65,75].

Overall, vortioxetine-induced synaptic plasticity and cognitive improvement may be attributable to vortioxetine’s effects on various types of 5-HT receptors. The blockade of ionotropic 5-HT3 receptors appears to play a prominent role in its mechanism of action.

## 5. Glutamate Receptor Antagonists (GRAs)—Ketamine/Esketamine, Dextromethorphan-Bupropion

### 5.1. Ketamine and Esketamine

The discovery of the rapid antidepressant and antisuicidal effects of ketamine [76,77,78] is perhaps the most significant advancement in managing MDD since the approval of the first effective antidepressant, imipramine, in the mid-1950s. More importantly, ketamine is the first antidepressant to introduce a paradigm shift from a monoamine to glutamatergic hypothesis [79].

The primary mechanism of action of ketamine is the non-competitive antagonist of N-methyl-D-aspartate (NMDA) receptors (Table 1). However, the rapid onset of antidepressant effects of ketamine is not directly associated with the NMDA receptor blockade. Instead, there may be a compensatory increase in glutamate release to overcome the reduced availability of NMDA receptors. Increased glutamate, as a result, is available to activate other glutamate receptors, such as AMPA receptors, increasing brain-derived neurotrophic factor (BDNF) [80]. This increase in BDNF promotes neuroplasticity in brain areas involved in MDD [81]. In addition, low-dose ketamine may have neuroprotective effects via inhibiting proinflammatory cytokines, such as interleukin-6 and tumor necrosis factor-Alpha [82]. However, since ketamine has agonist effects on mu-opioid receptors, concerns have been raised about the addiction potential of ketamine. A small sample study showed the absence of antidepressant effects of ketamine after mu-opioid receptors were blocked with naltrexone [83]. Not only were these study findings challenged in a letter to the editor [84], but, to our knowledge, no other study has replicated these results. On the contrary, the glutamatergic hypothesis of depression has provided new insights into developing more rapidly acting and sustainable antidepressants.

The early evidence of ketamine’s efficacy was based on two small sample studies [85,86], which reported a significant reduction in treatment-refractory symptoms after intravenous ketamine was added to the ongoing treatment. The results from these pilot studies were promising enough to warrant further augmentation trials in patients with TRD that replicated earlier findings [76,87]. However, it was the S-enantiomer of ketamine, esketamine, that underwent preclinical trials to successfully receive FDA approval, mainly due to aits easier route of administration (i.e., intranasally) and potentially fewer adverse effects [88,89]. Preclinical data provide significant evidence to support the rapid initiation of antidepressant effects as an augmentation treatment in patients with TRD [89,90,91]. However, in one preclinical study, the esketamine effects did not differ from placebo in older adults [88]. Nevertheless, based on a post hoc analysis, a significant response was reported in patients between 65–75 years old but not in those who were ≥75 years of age [88]. Since esketamine effects may be mediated by increased synaptic plasticity, the age-related changes in synaptic plasticity may explain the lack of response in older subjects. Although the rapid onset of esketamine efficacy may be secondary to that of dissociative symptoms, the preclinical data did not support this notion [92]. However, since esketamine has psychedelic effects, there is an abuse potential due to altered sensory perceptions, which may be one of the reasons. Its role as a dissociative agent with abuse potential may be why the FDA requires registration with risk management and mitigation services (REMS) before pharmacies can dispense esketmaine intranasal spray to the psychiatric facility/clinic administered under clinical supervision with a two-hour monitoring period for safety reasons.

Multiple CYP enzymes, including CYP2B6, CYP3A4, CYP2C9, and CYP2A6, metabolize esketamine, which increases the risk of interactions with multiple drugs that are substrates for these enzymes [3] (Table 1). For example, ketoconazole has been shown to increase esketamine levels, and rifampicin and St. John’s wort have been shown to lower esketamine levels [3]. Equally important are the genetic polymorphisms in CYP enzymes metabolizing esketamine, particularly CYP2D6, and CYP2B6, which can also alter esketamine levels [3].

### 5.2. Dextromethorphan–Bupropion

Dextromethorphan has been used as an antitussive agent for decades, but relatively recently, it has been found to have a complex pharmacology. Although it is typically classified as an opioid, it lacks the addictive and analgesic effects of opioids. Instead, it has other mechanisms of action, including serotonin agonism and subanesthetic blockade of NMDA receptors to mediate antidepressant and other psychotropic effects [4]. Since significantly high dextromethorphan levels are required for NMDA receptor blockade, dextromethorphan has been combined with an older antidepressant, bupropion, that inhibits dextromethorphan’s metabolism [4]. This is an excellent example of how pharmacokinetic drug–drug interactions can be effectively used to optimize a drug’s response. Since emotional dysregulation due to neurological disorders involves glutamate dysfunction, NMDA receptor blockade with dextromethorphan has also been successfully used to manage pseudobulbar affect. However, in this case, dextromethorphan was combined with quinidine instead of bupropion, which also increases dextromethorphan’s levels [93]. The dextromethorphan and bupropion combination was designated as a breakthrough treatment for MDD by the FDA due to its promising results [94]. The recent approval of dextromethorphan–bupropion was based on rapid and more significant response and remission rates being observed as early as in week 1, compared to those of the active control, bupropion monotherapy, which is consistent with rapid onset of antidepressant effects with ketamine, another NMDA receptor blockade [95]. One tablet of the combination treatment includes 45 mg of dextromethorphan and 105 mg of bupropion, which can be increased by taking it twice daily after three days of its initiation. The combination was found to be safe, and the adverse effects were similar between the combination and control groups except for dizziness, which was significantly higher in the dextromethorphan–bupropion group than in the control group with similar rates of discontinuation in the two treatment groups [4]. However, since bupropion lowers the seizure threshold, clinicians must rule out electrolyte imbalance due to any medical condition, such as vomiting, diarrhea, or an eating disorder, seizure history, and space-occupying brain mass, before using this antidepressant [96]. Concomitant use with drugs that also lower the seizure threshold requires monitoring as well. In addition, blood pressure should be monitored to avoid hypertension [97]. Finally, since bupropion inhibits CYP2D6, caution is required in concomitantly using drugs that are substrates for CYP2D6 [98,99]. Similar precautions are warranted in patients who have genetically mediated changes in CYP2D6 activity. In this context, the poor metabolizers of CYP2D6 may require lower doses and ultra-rapid metabolizers may require higher doses of dextromethorphan–bupropion [4].

In the short-term preclinical trials, dextromethorphan–bupropion was not associated with weight gain or sexual dysfunction, perhaps due to the noradrenergic effects of bupropion [94,100,101]. In addition, dextromethorphan–bupropion did not produce psychotomimetic effects, possibly due to a transient blockade of NMDA receptors [94,100,101]. Overall, dextromethorphan–bupropion offers another option to manage depressive symptoms in nonrefractory patients and, unlike esketamine, can be administered orally.

## 6. GABA Positive Allosteric Modulator (GPAM)—Brexanolone

Brexanolone is a positive allosteric modulator (PAM) of gamma amino butyric acid-type A (GABA-A) receptors and is the first FDA-approved drug to manage postpartum depression (PPD) [102,103]. Although the neurobiological underpinnings of PPD are highly complex and not fully understood, several studies have reported GABA deficits underlying PPD [104]. Thus, the approval of a GABAergic drug to manage depressive symptoms supports the GABAergic hypothesis of depression [2,105]. However, since brexanolone’s administration takes about 60 h for infusion and is extremely expensive, another oral analog of allopregnanolone, zuranolone, is currently under review for FDA approval to treat MDD and potentially PPD [5]. Like esketamine, a brexanolone prescription also requires risk evaluation and mitigation strategy (REMS) registration to monitor potentially serious adverse effects that could occur with brexanolone infusion [106].

Based on some basic research, brexanolone appears to have multiple effects that may be beneficial for improving PPD. The role of GABA in depression has been supported by the altered composition of GABA-A receptor subunits in depressed patients [2,105]. Brexanolone enhances the inhibitory effects of GABA-A receptors and stabilizes allopregnanolone (a progesterone metabolite) during and after pregnancy by opening chloride channels to hyperpolarize and suppress neural activity in the brain [107,108]. Thus, stimulation of GABA-A receptors results in anxiolytic and antidepressant effects [103,109]. Other GABAergic drugs, such as fengabine, have also been shown to improve depressive symptoms in animal models of depression [110]. In addition, studies have also reported the compromised role of GABA-A receptors in modifying the stress response and reducing depressive symptoms, which may play a part in PPD [110]. GABAergic neurotransmission also controls hippocampal neurogenesis and neural maturation [2], explaining the cognitive and memory deficits observed in depressed patients. In addition, progesterone and its primary metabolite, allopregnanolone, directly regulate the biological mechanisms involved in emotion processing and cognition and the neural reward system involved in depression [111]. Additional studies have associated decreased levels of allopregnanolone with an increased presence of depressive symptoms in pregnant women [112], and increased allopregnanolone levels have been linked to a lower risk of PPD [113].

Preclinical data support brexanolone’s effectiveness by showing that a more significant response and remission rate was seen with brexanolone infusion versus placebo treatment, which was maintained throughout the 30-day follow-up period [114,115]. Notably, only four out of ten study subjects experienced adverse effects in the brexanolone group compared to the eight out of eleven subjects experiencing these in the placebo group. There were no mortalities or serious adverse events in any of the study participants. Common adverse effects included dizziness and sedation. In addition, some patients reported mild to moderate infusion site discomfort, pain, erythema, and an increase in thyroid-stimulating hormone, flushing, and oropharyngeal pain [114,115]. In addition, brexanolone treatment is not recommended for patients with renal dysfunction and should be avoided by patients using benzodiazepines to prevent excessive sedation and respiratory depression [116].

In terms of drug–drug interactions, brexanolone can inhibit CYP2C9, affecting the metabolism of drugs that are substrates for CYP29, such as tolbutamide and S-warfarin [115]. However, unlike the majority of other antidepressants that are primarily metabolized by the phase-I enzymes (i.e., CYP enzymes), brexanolone undergoes phase-II metabolism, including ketoreduction, glucuronidation, and sulfation, and is not likely to be affected by the concurrent use of drugs that inhibit CYP enzymes [114,116,117]. However, drug-induced alterations in the activity of phase II enzymes metabolizing brexanolone may compromise its efficacy and/or tolerability, though pharmacokinetic drug interactions involving brexanolone have not been studied formally. Similar results may be produced by genetic polymorphisms in phase-II enzymes metabolizing brexanolone.

## 7. Conclusions

For the last seven decades, antidepressants have been developed based on the monoamine hypothesis, which involves the modulation of one or more of the three monoamines, dopamine, norepinephrine, and serotonin. However, recent paradigm shifts from monoamine to glutamatergic and GABAergic hypotheses have addressed some of the unmet needs in treating depression. Although the prototype antidepressants with glutamatergic and GABAergic mechanisms are expensive and require mandatory monitoring, relatively cheaper and easier delivery systems have been FDA-approved or are undergoing phase-III trials, such as the orally administered dextromethorphan–bupropion combination with ketamine-like effects and zuranolone with brexanolone-like effects. However, the bench-to-bedside application of the latest developments in antidepressants remains compromised, which is primarily due to the widening gap between the explosive increase in neuroscientific knowledge and its practical application during psychopharmacology training. Perhaps, more importantly, the high cost of the newer antidepressants makes it extremely difficult for the most challenged depressed patients from a low socioeconomic class to access these medications.

## Figures and Tables

**Table 1 jpm-13-00773-t001:** Clinical and pharmacological characteristics of newer antidepressants.

Name	Class	Indication	Half-Life Hours	Dose mg/day	Route	PK Drug-Drug Interactions	Gene-Drug Interactions	Molecular Targets	Clinical Utility	Common AEs
Vilazodone	SPARI	Adult MDD	25	40	Oral	Concurrent use of CYP3A4 substrates, Inhibitors, or Inducers	CYP3A4 polymorphisms are clinically insignificant	-SERT, inhibitor -5HT1A receptor partial agonist	Compared to SSRIs: -Lower sexual dysfunction -Lesser initial anxiety -Lesser emotional blunting -Faster onset of efficacy, -Lower risk for long-term loss or decrease in efficacy	Frequent GI upset, dizziness, insomnia, fatigue, jitteriness
Levomilnacipran	SNRI	Adult MDD & Fibromyalgia	12	40–120	Oral	Concurrent use of CYP3A4 substrates, Inhibitors, or Inducers	CYP3A4 polymorphisms are clinically insignificant	-NET inhibition more than SET inhibition	Activating, less emotional and cognitive blunting, improves fatigue, low weight gain,	Nausea, headache, dry mouth, hyperhidrosis, tachycardia, hypertension
Vortioxetine	MMA	Adult MDD	57	5–20	Oral	Concurrent use of CYP2D6 substrates or Inhibitors	CYP2D6 polymorphisms are clinically significant	-5-HT3, 5HT7, & 5HT1D receptor antagonist, -5-HT1B receptor partial agonist -5-HT1A receptor agonist -SERT inhibitor	Minimal sleep effects, improved psychomotor speed, fewer discontinuation symptoms	Less GI upset, minimal sexual dysfunction
Esketamine	GRA	-Adult TRD -Augmentation treatment	7–12	-Day 1—56 mg -Wk. 1–8—56 or 84 mg twice/wk. -Wk. 9—56 or 84 mg once or twice/wk.	Intranasal spray	Concurrent use of CYP2A6, CYP2B6, CYP2C9, or CYP3A4 substrates, inhibitors, or inducers	Polymorphisms in CYP2B6 can be clinically significant	-NMDA receptor antagonism, -mu-opioid receptor blockade	Rapid antidepressant and antisuicidal effects	Dissociation, dizziness, nausea, sleepiness, vertigo, headache, dysgeusia, numbness, anxiety, flushing, hypertension
Dextromethorphan (DXM) -Bupropion (BUP)	GRA	Adult MDD	22	DXM = 45 mg plus BUP = 105 mg. 1 tab once/day × 3 days, then -1 tab two times a day	Oral	Concurrent use of CYP2D6 substrates or inhibtors for DXM Concurrent use of CYP2B6, substrates, inhibitors, or inducers for BUP	Polymorphisms in CYP2D6 and CYP2B6 are clinically significant	-DXM - NMDA receptor antagonism -BUP – NE and DA reuptake pump blockade	Rapid antidepressant effects	Anxiety, psychosis, hypomania, confusion, decreased concentration, seizures, hypertension
Brexanolone	GPAM	PPD	9	90 mcg/kg/h	60-h IV infusion	Concurrent use of substrate inhibitors, or inducers of phase II enzymes ketoreductase, Glucuronyl transferase, and sulfatase	Polymorphisms of ketoreductase, glucuronyl transferase, and sulfatase)	-PAM for GABA-A receptors	Rapid antidepressant effects	Sedation, injection site discomfort/erythema, pain/rash, dizziness, flushing, oropharyngeal pain, increased TSH, loss of or altered consciousness

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
