# Peer review of "Recent Developments in Pharmacotherapy of Depression: Bench to Bedside"

_jpm, 2023, doi:10.3390/jpm13050773_

Round 1

Reviewer 1 Report

This is an interesting review paper on antidepressants, from traditional to recently developed molecules. The manuscript is well written and clear, focusing on recent advances in the pharmacotherapy of depression, including relatively new monoamine antidepressants, vilazodone, a selective partial agonist and reuptake inhibitor(SPARI), levomilnacipran, the latest and unique addition to the serotonin-norepinephrine reuptake inhibitors (SNRIs), and vortioxetine, a multimodal antidepressant (MMA). Furthermore, glutamate receptor antagonists and GABA positive allosteric modulators were discussed appropriately, with their  advantages and limitations.

The review covers almost all important molecules in the field, with an emphasis on treatment-refractory depression. There are only a few suggestions/concerns to improve the manuscript structure and content: 

- I miss some illustrations, such as tables or graphs, for better visibility

- There are almost no pharmacogenetics information related to antidepressants, which is very important aspect for the current and future practice; there are numerous data available in the literature, e.g. "Pharmacogenetics of antidepressants: A step to individualized therapy", doi: 10.5937/SMCLK2001013S

Minor:

A list of abbreviations at the end of the paper would help with reading and interpretation

Reviewer 2 Report

This manuscript consists of an overview of different antidepressants that have been approved quite recently (or are undergoing clinical trials) and is mainly focused on limitations about the low practical use of them. Their mechanism of action is described as well as potential interactions with other drugs. Different paradigms are explored and they are used as justification of the antidepressant effect of the analysed drugs. The topic is relevant since these drugs are not widely used and possible causes are suggested for each one. The review includes many references and each drug is explained systematically.

There are some aspects that could be improved in the paper:

1)      At present, pharmacogenetics adds interesting information about the suitability of drugs for treatment of depression within personalized medicine. Information about this aspect should be included in the review.

2)      It is suggested to include a summary as a table with advantages/disadvantages for each drug in order to be useful for physicians.

3)      Metabolizing profile of ketamine & esketamine is not described.

4)      Minor change: final of parenthesis (line 138)

5)      Minor change: number after (SNRI) (line 157)

6)      Minor change: its instead it’s (line 160)

7)      Minor change: underline (line 179)

8)      Minor change: different size of letter CYP3A4 (lines 206 to 218)

9)      Minor change: different size of letter (line 285)

10)   Minor change: CYP2C9 is not well written (line 138)

Round 2

Reviewer 2 Report

The paper has been changed according to given recommendations.